# UMC: Unified Malfunction Controller for Damage-Resilient Legged Locomotion

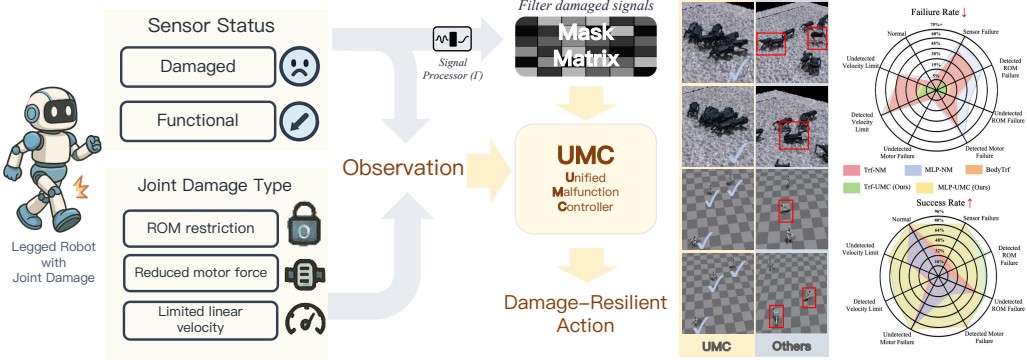

(a) From Malfunctions to Robust Actions  (b) Comparison

Figure 1: (a) We consider diverse malfunction scenarios in legged robots, including different sensor statuses and three types of joint damage. These malfunctions are reflected in the observation input. Two-stage pipeline UMC employs a masking mechanism to filter corrupted signals and produces damage-resilient actions. (b) Comparison with existing approaches. UMC achieves lower failure rates and higher 3 unit performance across various malfunction scenarios, as illustrated by both qualitative rollouts (red boxes highlight failure cases of baselines) and quantitative radar plots.

## Abstract

Adaptation to unpredictable damages is crucial for autonomous legged robots, yet existing methods based on multi-policy or meta-learning frameworks face challenges like limited generalization and maintenance. In this work, we first provide a systematic categorization of eight representative malfunction scenarios, covering both detectable and undetectable cases. Then, we propose a novel model-free framework, **U**nified **M**alfunction **C**ontroller (UMC). UMC employs a two-stage training pipeline: the first stage learns strong baseline locomotion in undamaged environments, while the second stage fine-tunes the controller with mixed malfunction scenarios to encourage adaptive and robust behavior. For detectable settings, we introduce a masking strategy that explicitly filters corrupted signals, preventing error propagation and enabling policies to rely on functional joints. UMC is compatible with both transformer and MLP backbones. Experiments across multiple humanoid and quadruped tasks show that UMC consistently reduces failure rates and improves task completion under diverse damage conditions. The source code and trained models will be made available to the public.

## 1 Introduction

Legged robots have achieved remarkable progress due to their flexibility and adaptability across diverse scenarios. Prior research has mainly focused on network design, leveraging observational signals and proprioceptive states. However, robustness to diverse scenarios remains unexplored, which is critical when joints or limbs malfunction, especially when human intervention is impractical or even impossible (Hutter et al., 2017; Bellicoso et al., 2018; Wensing et al., 2022). For example, in disaster recovery, a search-and-rescue robot navigating through rubble may suffer joint

failures caused by debris, making external assistance unsafe or impractical. Therefore, robustness is essential for reliable real-world deployment.

Existing approaches address this challenge via multi-policy frameworks, model predictive control, data augmentation (Kume et al., 2017; Yang et al., 2021; Hou et al., 2024; Mayne et al., 2005), or meta-learning (Nagabandi et al., 2019; Raileanu et al., 2020; Guo et al., 2023; Chen et al., 2024). However, they often suffer from complex maintenance, limited generalization, and degraded performance under out-of-distribution deployment. Recently, (Skand et al., 2024) present a masking strategy that appends binary indicators of sensor failure to proprioceptive inputs, but it yields only marginal gains. As shown in Fig. 1(b), methods trained under specific damage settings exhibit high failure rates for both humanoid and quadruped robots in open-world tests.

To overcome these limitations, we take a systematic perspective on malfunction modeling and controller design (Fig. 1(a)). First, we analyze multiple malfunction scenarios and categorize them into eight types in Table 1, covering most practical cases. These both detectable and undetectable malfunctions, characterized by three key parameters: position, velocity, and motor force, which manifest as restricted range of motion, reduced motor force, and limited linear velocity. During training, malfunctions are randomly applied to different joints, encouraging the model to generalize across diverse combinations.

To this end, we propose a **U**nified **M**alfunction **C**ontroller (UMC), with a powerful *two-stage training pipeline*. Concretely, in **Stage I**, the model is trained under undamaged environments to acquire strong ability under normal conditions. This step provides a stable initialization and avoids the instability that often arises when training directly under diverse malfunctions. In **Stage II**, the pretrained model is fine-tuned with a mixture of malfunction scenarios, which can be regarded as structured noise injection. This adaptation forces the policy to generalize across diverse impairments while preserving the capabilities learned in Stage I.

For undetectable malfunctions, sensors remain functional while joints are impaired, and the network adapt autonomously to different types of joint damage, improving robustness and generalization under challenging conditions. For detectable malfunctions, sensor damage invalidates signals from affected joints, posing greater challenges since the controller must operate without reliable feedback. To handle this, we introduce a masking strategy that leverages the known malfunction information to explicitly filter out the faulty inputs, preventing the network from being misled and enabling it to exploit the remaining reliable signals. Our framework supports both transformer- and MLP-based action networks, enabling effective adaptation under diverse failures.

Compared to previous methods, our approach reduces the average fail rates by 30% and 37% with the base structure of the transformer and the MLP on three locomotion tasks, as shown in Fig. 1(b). The main contributions are:

- We systematically analyze and categorize eight representative malfunction scenarios, including detectable and undetectable malfunctions based on sensor status, all characterized around the three key joint parameters: position, velocity, and motor force.

- We propose the Unified Malfunction Controller (UMC), a powerful two-stage training pipeline. The first stage ensures strong baseline capabilities under normal conditions, while the second stage fine-tunes the model under diverse damage scenarios to enhance robustness and adaptability. For detectable malfunctions, we further introduce a masking strategy that leverages explicit malfunction information to reduce the influence of damaged joints on the network.

- We conduct extensive experiments on three locomotion tasks with both transformer-based and MLP-based action networks, demonstrating that our method significantly reduces failure rates and generalizes effectively across all eight damage scenarios.

## 2 RELATED WORK

**Reinforcement Learning in Legged Robots.** In recent years, reinforcement learning (RL) has gained traction in legged robots' control and locomotion tasks (Strudel et al., 2020; Tang et al., 2020). Some deep-learning-based RL methods are proposed to improve quadrupedal robots' stability across diverse terrains through combined simulated and real-world training (A & Jisha, 2022). In

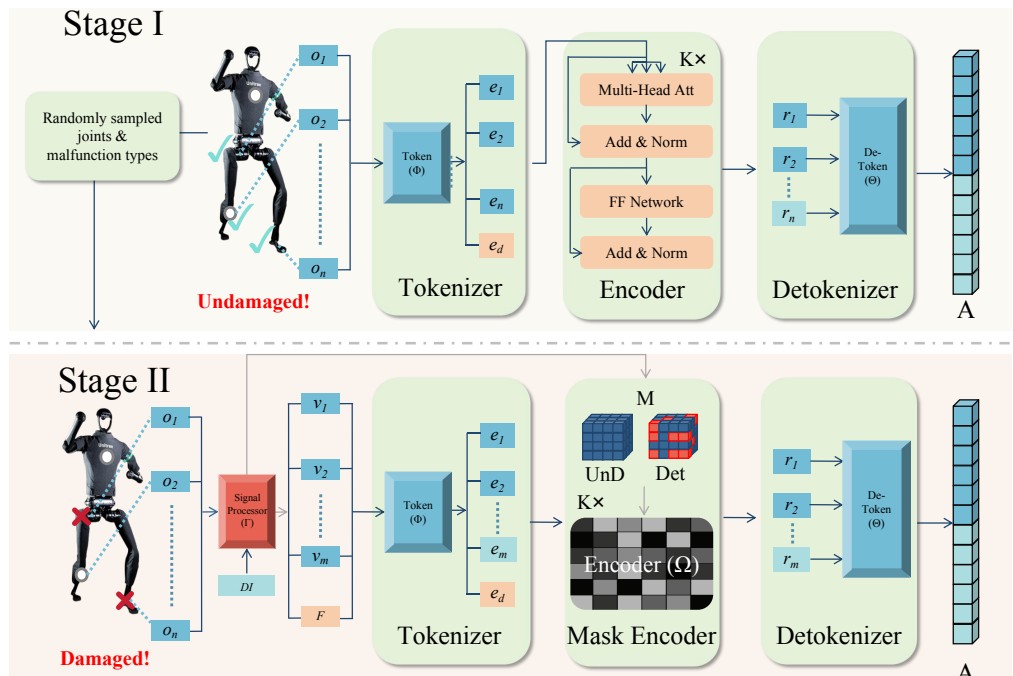

Figure 2: UMC pipeline (Transformer-based): a two-stage policy for damage-resilient legged control. $\Gamma$ derives input-level masked observations $V$ and an attention mask $M$ from raw proprioception $O$ (plus a global flag $F$). The tokenizer forms $V$ and projects to $E$; a $K$-layer mask encoder injects $M$ into attention logits; the detokenizer maps tokens to actions $A$.

this work, we utilize the Proximal Policy Optimization (PPO) algorithm provided by Legged Gym for RL-based control of legged robots. Our approach focuses on enhancing locomotion control by reconstructing the Actor model, improving performance in complex environments.

**Self-recovering Robots.** In recent years, self-recovering robots have attracted significant interest in robotics research (Kawabata et al., 2002; Guan et al., 2015; Nwaonumah & Samanta, 2020). As robotics technology matures, enabling legged robots to adapt to joint damage has become increasingly critical. However, few studies address this directly, and existing approaches often lack generalization, require excessive training data, or complex maintenance with conflicting strategies (Kume et al., 2017; Nagabandi et al., 2019; Raileanu et al., 2020; Yang et al., 2021; Chen et al., 2022; Guo et al., 2023). Therefore, we aim to handle a broad spectrum of damage conditions with a single, unified policy, offering insights for future work.

**Transformer Models in Robotics** Transformers have gained sufficient popularity in various domains, including natural language processing (Vaswani et al., 2023), computer vision (Dosovitskiy et al., 2021), and are now being explored in robotic control due to their ability to model sequential dependencies and capture complex long-range relationships in data. Recent studies have also demonstrated the effectiveness of transformer-based architectures in the robotics field (Chen et al., 2021; Kurin et al., 2021; Gupta et al., 2022; Hong et al., 2022; Radosavovic et al., 2024). While inspired by BodyTransformer (Sferrazza et al., 2024), our work pursues a different objective—robustness to damage. Consequently, BodyTransformer performs poorly under failure conditions, whereas our approach achieves substantially better robustness.

## 3 PROPOSED METHOD

We aim to design a unified policy that enables the robot to complete tasks within various damage conditions. We explore the robustness of legged robots by systematically analyzing various damage factors and proposing a two-stage unified malfunction controller (UMC) framework to address them. In the following, we first specify the malfunction scope in Sec. 3.1, then introduce the UMC

framework and its key components in Sec. 3.2, and finally present the two-stage training pipeline and its generalization to MLP backbones in Sec. 3.3.

### 3.1 MALFUNCTION SETTINGS

We consider eight damage scenarios spanning both sensor failures and joint impairments, covering a broad range of realworld conditions (Table 1). This extends prior work that focused only on self-diagnosis or limited damage types (Guan et al., 2015; Quamar & Nasir, 2024). For sensors, we define two statuses: ***Damaged***, where they fail to provide correct observation and output only zeros (thus detectable); ***Functional***, where sensors provide accurate observation readings.

Table 1: Eight Damage Scenarios for Legged Robots.

| Scenario | Sensor Status | Joint Damage Type |
|---|---|---|
| 1 | Functional | None |
| 2 | Functional | ROM Restriction |
| 3 | Functional | Reduced Motor Force |
| 4 | Functional | Limited Linear Velocity |
| 5 | Damaged | None |
| 6 | Damaged | ROM Restriction |
| 7 | Damaged | Reduced Motor Force |
| 8 | Damaged | Limited Linear Velocity |

We consider three categories of joint damage: **range-of-motion (ROM) restriction**, **reduced motor force**, and **limited linear velocity**. ROM restriction: the joint cannot traverse its normal range (e.g., mechanical obstruction or controller-imposed limits). Reduced motor force: the available torque/force output is diminished (e.g., wear or partial failure). Limited linear velocity: the joint speed is capped, often by thermal protection or safety constraints. Further details of our malfunction specification are provided in the Appendix B.

### 3.2 UMC FRAMEWORK

**Problem Formulation.** We cast damage-robust legged control as a Markov Decision Process (MDP) $\mathcal{M} = (\mathcal{S}, \mathcal{A}, \mathcal{P}, r, \gamma)$. At time $t$, the agent observes a state $s_t \in \mathcal{S}$ built from per-joint signals $O_t = \{(q_i, \dot{q}_i, a_{i,t-1})\}_{i=1}^N \in \mathbb{R}^{N \times 3}$, where $q$, $\dot{q}$ and $a$ are the joint's position, velocity and previous action. For 1-DoF (revolute) joints, the action space is $\mathcal{A} = \mathbb{R}^N$, and the policy outputs one scalar command per revolute joint, $a_t = \pi_\theta(s_t) \in \mathbb{R}^N$ (e.g., torque/position/velocity targets). The environment transition is governed by the robot dynamics $s_{t+1} \sim \mathcal{P}(\cdot \mid s_t, a_t)$, and the agent receives a task reward $r_t = r(s_t, a_t)$; we optimize the standard discounted return $\sum_{t=0}^T \gamma^t r_t$.

We train an actor–critic with Proximal Policy Optimization (PPO). The critic $V_\psi(s_t)$ shares the base architecture with the actor but is used only during training to estimate returns/advantages. At inference, we deploy the actor alone to produce action sequences.

#### 3.2.1 TWO-STAGE TRAINING PIPELINE

As shown in Fig. 2, we adopt a two-stage training pipeline to ensure robustness in both normal and damaged settings.

**Stage I.** The model is trained under normal conditions to acquire strong baseline capabilities, which provides a solid foundation for subsequent adaptation.

**Stage II.** Building on the pretrained baseline, we fine-tune the model under diverse malfunction conditions to enhance generalization. This stage can be viewed as injecting structured noise into the system, forcing the model to learn robustness against potential failures. We consider two major categories of malfunctions:

1. **Undetectable malfunctions**: Scenarios 2-4 in Table 1, where the sensors remain fully functional but the joints themselves are impaired. In these cases, since the sensors remain operational, no masking is applied, and the model is directly fine-tuned to adapt its policy to the altered dynamics.

2. **Detectable malfunctions**: Scenarios 5-8 in Table 1, which simulates partial limb damage by adding joint and sensor damage to certain joints and allowing the detection of damage information. To prevent corrupted observations from interfering with decision making, we introduce a masking strategy that explicitly suppresses signals from failed joints.

During Stage II, detectable and undetectable malfunctions are mixed together with normal condition (Senario 1 in Table 1) for training. This joint training scheme prevents the model from overfitting to a single failure type and further improves its robustness across diverse scenarios.

### 3.2.2 ACTOR MODEL

we take the transformer structure as the base architecture for the UMC. For the actor model, as shown in Fig. 2, it consists of three main components: tokenizer, encoder, and a detokenizer. The tokenizer and detokenizer perform the transformation between the joint observation, a sequence of tokens and the action sequence, enabling seamless encoding and decoding processes. For damaged conditions, a masking strategy with mask encoder are designed to capture dependencies and refine input representations using only the embeddings of well-functioning joints.

**Base Structure.** We use a vanilla Transformer with a tokenizer $\Phi$, a encoder $\Omega$, and a detokenizer $\Theta$. Given $O \in \mathbb{R}^{N \times 3}$, the tokenizer applies a joint-wise linear projection to hidden size $D$ and adds learnable positional embeddings, producing

$$E = \Phi(O), \qquad E \in \mathbb{R}^{N \times D}. \tag{1}$$

The encoder $\Omega$ consists of several stacked attention blocks where each block has a multi-head self-attention and feed-forward network module. It outputs

$$R = \Omega(E), \qquad R \in \mathbb{R}^{N \times D}. \tag{2}$$

Finally, the detokenizer $\Theta$ applies joint-specific linear heads to the first $N$ tokens (the flag token is context-only) to produce actions:

$$A = \Theta(R_{1:N}), \qquad A \in \mathbb{R}^{N \times 1}. \tag{3}$$

**Masking Strategy.** For detectable malfunctions, the damage information (*DI*) is provided as part of the input, i.e., we explicitly know which joints are non-functional. Among the $N$ joints, only $M$ remain functional. We construct

$$V, M = \Gamma(O, DI). \tag{4}$$

where $V = \{v_i\}_{i=1}^{M+1} \in \mathbb{R}^{(M+1) \times 3}$, the first $M$ rows correspond to the observations of functional joints, and the last row is a global flag token $F \in \{-1, 1\}^{1 \times 3}$, with $F = [-1, -1, -1]$ if no malfunction is present and $F = [1, 1, 1]$ otherwise. This design provide the model with explicit contextual information about the presence of damage.

The masking matrix $M \in \mathbb{R}^{(N+1) \times (N+1)}$ encodes joint malfunction information and is injected into the self-attention module. Here, $N$ denotes the number of joint embeddings in $E$, and the additional $+1$ token corresponds to the global damage flag token. For damaged joints, the entries in $M$ are set to $-\infty$, forcing the attention weights of these joints to zero after softmax and thereby eliminating their influence. Formally, a self-attention layer computes

$$\text{Output} = \text{Softmax}\left(\frac{\mathbf{Q}(E)\,\mathbf{K}(E)^\top}{\sqrt{d_k}} + M\right)\mathbf{V}(E), \tag{5}$$

where $\mathbf{Q}(E)$, $\mathbf{K}(E)$, and $\mathbf{V}(E)$ are the query, key, and value projections of $E$, and $d_k$ is the dimension of the $\mathbf{Q}(\mathbf{E})$ and $\mathbf{K}(\mathbf{E})$. This design explicitly suppresses the contribution of malfunctioning joints, allowing the policy to focus on reliable signals.

**Training Loss.** The loss consists of actor and critic losses. Additionally, an entropy regularization term is included to promote exploration, which encourages the agent to maintain a diverse set of actions and avoid premature convergence to suboptimal policies. These components guide the optimization of both the policy and value functions.

The total loss function in PPO is defined as:

$$\mathbb{L} = \mathbb{L}_{\text{surrogate}} + \lambda_1 \cdot \mathbb{L}_{\text{value}} + \lambda_2 \cdot \mathbb{L}_{\text{entropy}}, \tag{6}$$

where $\lambda_1$ and $\lambda_2$ denote weight parameters. $\mathbb{L}_{\text{surrogate}}$, $\mathbb{L}_{\text{value}}$ and $\mathbb{L}_{\text{entropy}}$ are the loss of policy surrogate, value function, and entropy regularization, respectively. Please refer to our Appendix E for more details of the loss that are not the key points of our work.

| Methods | 1 unit ↑ | 2 unit ↑ | 3 unit ↑ | 4 unit ↑ | 5 unit ↑ | failed ↓ |
|---|---|---|---|---|---|---|
| Trf-NM | 81% | 64% | 54% | 46% | 38% | 7% |
| MLP-NM | 67% | 54% | 48% | 43% | 36% | 23% |
| BodyTrf | 84% | 68% | 56% | 45% | 36% | 10% |
| TFQL | 51% | 43% | 38% | 33% | 29% | 19% |
| MLP-UMC (Ours) | 93% | 88% | 83% | 78% | 66% | 4% |
| Trf-UMC (Ours) | **97%** | **95%** | **91%** | **84%** | **72%** | **2%** |

Table 2: Average Performance of Models on the A1 Task Across Eight Damage Conditions.

### 3.3 GENERALIZING UMC TO AN MLP BACKBONE

UMC is architecture-agnostic and can be instantiated with a MLP backbone. Unlike the Transformer variant, the MLP policy does not include self-attention and thus does not require tokenization or attention-bias masking.

We reuse the detector $\Gamma$ to produce the input-level masked observations $V$ and the global flag $F$ (Sec. 3.2.2): the attention-bias matrix $M$ is simply not used in this instantiation. We form $O' = \text{concat}(F, V) \in \mathbb{R}^{(N+1)\times 3}$ and flatten it to $O'' = \text{vec}(O') \in \mathbb{R}^{3(N+1)}$. The MLP then maps this vector to per-joint actions:

$$A = f_{\text{MLP}}(O''), \qquad A \in \mathbb{R}^{N\times 1},$$

where $f_{\text{MLP}}$ denotes an $L$-layer feed-forward network (e.g., GELU/ReLU activations with optional LayerNorm). This realization preserves UMC's input-level masking and global damage context while removing attention-specific components, demonstrating that the proposed masking strategy is orthogonal to the choice of backbone.

## 4 EXPERIMENTS

In this section, we begin by describing the experimental setup, followed by evaluation metrics. Next, we present both quantitative and qualitative comparison results with existing methods. Finally, we conduct extensive ablation studies to validate the effectiveness of the proposed model.

**Implementation Details.** All models are trained on a single Nvidia A6000 GPU and evaluated using PPO-based Reinforcement Learning (Schulman et al., 2017) for three different robot locomotion tasks, which are the A1-Walk task from ParkourGym (Zhuang et al., 2023) and the H1 and G1 tasks from Unitree. For SOTA work comparison, we selected the Solo8 task (Grimminger et al., 2020). Among them, A1 and Solo8 are quadruped robots, while H1 and G1 are humanoid robots. All these locomotion tasks are performed within the IsaacGym environment (Makoviychuk et al., 2021), managed by the Legged Gym Repository (Rudin et al., 2022). We provide transformer-based and MLP-based UMC architectures. Please refer to the Appendix A for more details on model configurations, malfunction limits, and other parameters.

**Damage Settings During Inference.** During inference, we apply three distinct damage settings for every task, all of which differ from those used during the training stage. First, for the rough terrain task A1, the robots operate on a terrain that is different from those encountered during training. Second, different joint combinations are randomly selected using various seeds to introduce damage. Third, malfunctions are introduced at different times during inference to simulate more different robot gaits when suffering damage and different combinations of joint damage, thereby preventing the model from completely relying on the prior knowledge learned from the training set. For example, in one environment, a robot may lift one of its front legs, whereas in another, the same leg may point downward when its corresponding joints are damaged.

**Evaluation Metrics** After legged robots walk certain steps under normal conditions, we apply damage to them and record the initial position. During the subsequent episodes, we record the following comprehensive metrics to validate the locomotion capabilities of legged robots.

Specifically, we evaluate whether the robots can move beyond the radii of 1, 2, 3, 4, and 5 units (0.5, 1, 1.5, 2, and 2.5 units for the Solo8 task) from their initial positions without falling. If the robot can maintain its original trajectory despite the damage, this distance should correlate positively

| Methods | 1 unit ↑ | 2 unit ↑ | 3 unit ↑ | 4 unit ↑ | 5 unit ↑ | failed ↓ |
|---|---|---|---|---|---|---|
| Trf-NM | 44% | 43% | 42% | 39% | 35% | 56% |
| MLP-NM | 33% | 33% | 31% | 29% | 25% | 67% |
| BodyTrf | 52% | 51% | 49% | 45% | 40% | 48% |
| MLP-UMC (Ours) | 86% | 85% | 80% | 73% | 64% | 14% |
| Trf-UMC (Ours) | **91%** | **90%** | **85%** | **79%** | **70%** | **9%** |

Table 3: Average Performance of Models on the G1 Task Across Eight Damage Conditions.

| Methods | 1 unit ↑ | 2 unit ↑ | 3 unit ↑ | 4 unit ↑ | 5 unit ↑ | failed ↓ |
|---|---|---|---|---|---|---|
| Trf-NM | 57% | 56% | 55% | 51% | 46% | 43% |
| MLP-NM | 57% | 57% | 55% | 52% | 47% | 43% |
| BodyTrf | 53% | 53% | 51% | 48% | 44% | 47% |
| MLP-UMC (Ours) | **97%** | **97%** | **94%** | **88%** | **80%** | **3%** |
| Trf-UMC (Ours) | 95% | 94% | 90% | 84% | 75% | 5% |

Table 4: Average Performance of Models on the H1 Task Across Eight Damage Conditions.

| Methods | 0.5 unit ↑ | 1 unit ↑ | 1.5 unit ↑ | 2 unit ↑ | 2.5 unit ↑ | failed ↓ |
|---|---|---|---|---|---|---|
| MT-FTC | 39% | 31% | 30% | 30% | 29% | 46% |
| Trf-UMC (Ours) | **73%** | **67%** | **60%** | **52%** | **41%** | **12%** |

Table 5: Average Performance of Models on the Solo8 Task Across Eight Damage Conditions. 'MT-FTC' is the method proposed in (Hou et al., 2024).

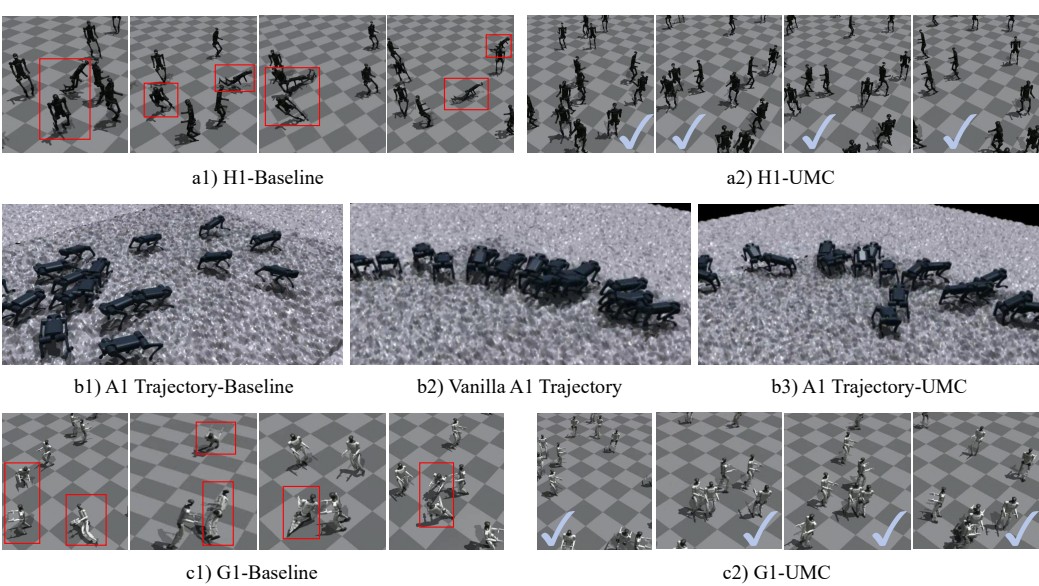

a1) H1-Baseline  a2) H1-UMC

b1) A1 Trajectory-Baseline  b2) Vanilla A1 Trajectory  b3) A1 Trajectory-UMC

c1) G1-Baseline  c2) G1-UMC

Figure 3: Qualitative Comparison Between Methods Under Damaged Scenarios. 'Baseline' refers to robots trained using baseline methods, while 'UMC' denotes robots trained with the UMC method. Figure 'b2)' shows a snapshot of the original trajectory at a specific time point under undamaged conditions, while b1) and b3) are in damaged conditions.

with time. Therefore, a greater distance travelled indicates a more effective policy, as it allows the robot to move further given the limited time. Additionally, legged robots that exhibit any falling behaviour are excluded from the previous distance statistics and are instead counted in a separate metric labelled as 'failed'.

| Ratios | 1 unit ↑ | 2 unit ↑ | 3 unit ↑ | 4 unit ↑ | 5 unit ↑ | failed ↓ |
|---|---|---|---|---|---|---|
| 1:1:1:0 | 89% | 84% | 79% | 73% | 66% | 10% |
| 1:1:0:1 | 90% | 83% | 75% | 65% | 50% | 2% |
| 1:0:1:1 | 97% | 94% | 88% | 80% | 68% | 2% |
| 0:1:1:1 | 97% | **95%** | 90% | 82% | 69% | **2%** |
| 1:2:2:1 | 97% | 94% | 89% | 81% | 68% | 2% |
| 1:3:3:1 | 97% | 94% | 87% | 77% | 66% | 2% |
| 1:1:1:1 (Ours) | **97%** | 94% | **90%** | **84%** | **74%** | 3% |

Table 6: Average Performance of Transformer-Based UMC with Different Stage-II Environment Settings. The ratios correspond to four training scenarios in Stage II from left to right: 'Undamaged', 'Sensor-only Damage', 'Detectable Joint Damage', and 'Undetectable Damage'.

## 4.1 EXPERIMENTAL RESUTLS

As shown in Tables 2, 3, and 4, we present the average performance for each task with our metrics. The averaged results for each model are computed by summing performance across eight damage scenarios and three inference settings, demonstrating the superiority of our UMC framework. More statistics are shown in the Appendix C.

The UMC significantly reduces the number of fall cases on the H1, G1, and A1 tasks, and performs better against the BodyTransformer. Compared to the normally trained transformer, our baseline, the transformer-based UMC achieves an average reduction in failure rates across eight types of damage by 5%, 38%, and 47% in tasks A1, H1, and G1, respectively. Similarly, MLP-based UMC demonstrates reductions of 19%, 40%, and 53%, respectively. UMC prompts robots to rely more on their functional limbs when dealing with various failures, thereby effectively reducing the impact of damaged joints on their actions.

For the task completion performance of the transformer architecture, taking the A1-Walk task as an example, UMC improves the robot's achievement rates across the 1-unit to 5-unit metrics by 16%, 31%, 37%, 38%, and 34%, respectively. For the MLP architecture, the robot also demonstrates improvements of 38%, 38%, 35%, 33%, and 29% on the H1 task. The results show that the proposed masking mechanism enables rapid adaptation to new types of damage without the need to switch to a new policy. Therefore, robots can respond to sudden damage more quickly and adjust their gait accordingly.

Fig. 3 indicates that UMC can handle various damage conditions and effectively maintains the intended trajectory, which demonstrates that UMC can reduce the impact of damages from another perspective. Moreover, Fig. 1(c) show that UMC retains and even slightly enhances the robot's performance under normal, undamaged conditions across three tasks. This improvement comes from the design of our two-stage pipeline, which ensures that the trained robots maintain their performance under normal conditions.

Compared to the existing method, MT-FTC (Hou et al., 2024) in the Solo8 task, Table 5 indicates that UMC achieves a 26.8% improvement across 0.5-2.5 unit and reduces the fall rate by 34% compared to MT-FTC. These results demonstrate that UMC exhibits greater flexibility in adapting to various conditions compared to existing methods. We also compare our method with the other approaches proposed by (Liu et al., 2024) and (Skand et al., 2024) in Table 2 (abbreviated as TFQL and SMS, respectively). The considerable performance degradation observed in TFQL is primarily due to its reliance on corrupted sensor feedback, which adversely impacts the controller's decision-making. Moreover, its reliance on history rollback further exacerbates this issue. We attribute the performance gap to our method's advantage over SMS in treating each joint's observation as an independent sequence, enabling the transformer to better learn inter-joint relationships and produce more reasonable actions.

Furthermore, Tables 2, 3, and 4 reveal that our baseline degrades less in humanoid robot tasks than in quadruped robot tasks as the metric increases from 1 to 5 units. We attribute this to the structural differences between humanoid robots and quadruped ones like A1. Unlike A1, which can easily remain upright and stable despite finding it difficult to move forward due to damage, humanoid robots face greater challenges in maintaining balance during movement.

| Paradigms | 1 unit ↑ | 2 unit ↑ | 3 unit ↑ | 4 unit ↑ | 5 unit ↑ | failed ↓ |
|---|---|---|---|---|---|---|
| Curriculum-Based | 97% | 93% | 88% | 80% | 67% | **1%** |
| RMA-Based | 95% | 92% | 88% | 82% | 72% | 4% |
| Stage-Based (Ours) | **97%** | **94%** | **90%** | **84%** | **74%** | 3% |

Table 7: Average performance of Transformer-based UMC under different training paradigms. Paradigms include a curriculum schedule, an RMA-based adaptation scheme, and our two-stage (stage-based) scheme.

## 4.2 ABLATION STUDY

In this section, we conduct ablation studies on training stages and sampling ratios. For more details and ablation studies on masking value and foundational paradigms, please refer to the Appendix D.

**Training Stage.** The results are shown in Fig. 4. The blue and green curves represent the two training stages in our method, while the orange curve shows the one-stage training, training solely on our Stage II damaged environments. The curves show that the one-stage setting eventually fails to converge within 2500 iterations in the G1 task, whereas the two-stage approach proves effective across all three tasks. We attribute this to the introduction of an overly complex training set in

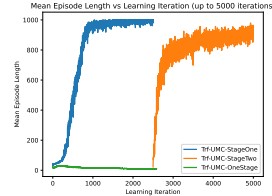 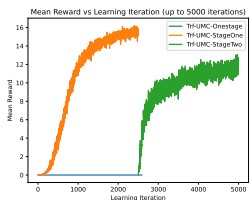

(a) Episode vs. Iteration  (b) Reward vs. Iteration

Figure 4: Comparison of One-Stage and Two-Stage Training in the G1 Task.

the initial stage, which hindered the model's convergence and ultimately prevented the discovery of effective policies. Thus, we select the two-stage training process for our workflow.

**Sampling Ratio.** We further divide 'undetectable damage' into 'sensor-only damage' (scenario 5) and 'detectable joint damage'(scenario 6-8). Table 6 demonstrates that the default ratio of 1:1:1:1 achieves the best overall performance. The potential reason is two-fold. First, unlike conditions that exclude certain subcategories (e.g., 1:1:1:0), the model could learn all four damaged scenarios with the default ratio during Stage II. Second, compared to ratios that focus more heavily on detectable damage (e.g. 1:2:2:1), the default ratio achieves better balance and thus enables the model to learn to handle various types of damage more comprehensively.

**Paradigms.** Table 7 compares the results of different training paradigms of UMC. Besides the stage-based pipeline, we evaluate an RMA-based (Kumar et al., 2021) paradigm and a curriculum learning strategy that gradually increases task difficulty (no damage to undetectable joint damage to detectable joint damage). Curriculum learning underperforms the stage-based method, as its progressive focus on harder tasks reduces adaptability to easier cases. The RMA-based paradigm also lags behind, since masking causes history rollbacks to lose critical information, making it difficult to fit a reliable latent vector for environment representation.

## 5 CONCLUSION

In this paper, we present UMC, a unified, model-free framework that substantially improves the resilience of legged robots facing various failure scenarios, including sensor malfunctions and joint issues such as restricted motion, weakened motor, or limited velocity. The proposed UMC adopts two training stages that enable fast adaptation to damaged conditions while maintaining good performance in undamaged normal states. Specifically, UMC incorporates a masking strategy, isolating faulty joints, allowing the robot to compensate by emphasizing unaffected limbs and adapting dynamically to diverse damage patterns. Experimental results validate that the proposed UMC consistently improves both transformer and MLP architectures across different robot models and tasks, markedly reducing failure rates and improving task success under variable damage conditions, further improving the adaptability of robotic systems in challenging environments.

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

## A    UMC Implementation Details

In this section, as shown in table 8, we provide detailed experimental parameters of our UMC structure to facilitate reproducibility and related operations.

## B    Malfunction and Experiment Settings

In this section, we use detailed statistics and Fig. 5 to illustrate our damage design further. We conducted three sets of tests for each task with different damage conditions. The training and inference parameters are provided in Table 9, Table 10, Table 11 and Table 12, where:

*Malfunction Timing* refers to the specific episode we apply malfunctions to the robots. *ROM Limit* indicates the range of motion for each joint in the environment is restricted to a certain percentage of its original range. *Motor Limit* specifies that the motor strength for each joint is capped at a certain value. *Velocity Limit* means the maximum speed of joint movement is a certain value. *Random Damage Range* denotes the number of randomly selected joints damaged in each environment. *Random Malfunction Seed* refers to the seed we use when randomly selecting which joints to be damaged for each environment. *Perlin Noise Seed*, *Track Width*, *Border Size* and *Track Block Length* emphasize that we test our methods on different terrain settings in Table 9.

During inference, each damage scenario is tested separately. Also, in each scenario, the malfunction limits (ROM, Motor and Velocity) are applied to the joints under three malfunction setting groups (the timing to add malfunction, different damage range, etc.). This approach ensures that the robot's limbs encounter a wide range of states, enhancing the robustness and rigour of the process. The rationale is that the difficulty of overcoming obstacles and completing tasks significantly depends on the robot's posture. For example, a malfunction occurring when a limb is fully extended to support the robot's weight presents a greater challenge compared to when the limb is retracted during a recovery phase. Therefore, we eventually set up various inference groups with different damage settings to generate as many postures as possible.

Table 8: Detailed Parameters of the transformer-based and MLP-based Actor Model.

| Parameter | MLP | Transformer |
|---|---|---|
| Stage-One Epochs | 2500 | 2500 |
| Stage-Two Epochs | 2500 | 2500 |
| Encoder Layers | 4 | 4 |
| Embedding Input Size | N/A | 120 |
| Feedforward Size | [256,512,256,256] | 128 |
| Attention Heads | N/A | 2 |
| Total Parameters | 345,100 | 366,164 |

Table 9: Malfunction Settings for Training and Inference in the A1-Walk Task.

(a) Training Parameters.

| Parameter | Values |
|---|---|
| Num Envs | 7400 |
| Random Damage Range | [2,4] |
| ROM Limit | Random 30% |
| Motor Limit | 5 |
| Velocity Limit | 3 |
| Track Width | 1.6 |
| Track Block Length | 2.0 |
| Border Size | 8 |
| Perlin Noise Seed | 1 |
| Random Malfunction Seed | 42 |
| Episode Length | 1000 |
| Malfunction Timing | N/A |

(b) Inference Parameters.

| Parameter | Values |
|---|---|
| Num Envs | 4000 |
| Random Damage Range | [4,5] |
| ROM Limit | Random 10% |
| Motor Limit | 8 |
| Velocity Limit | 3 |
| Track Width | 6.0 |
| Track Block Length | 6.0 |
| Border Size | 4 |
| Perlin Noise Seed | [100, 25, 75] |
| Random Malfunction Seed | [1, 800, 50] |
| Malfunction Timing | [75, 100, 125] |
| Episode Length | 750 |

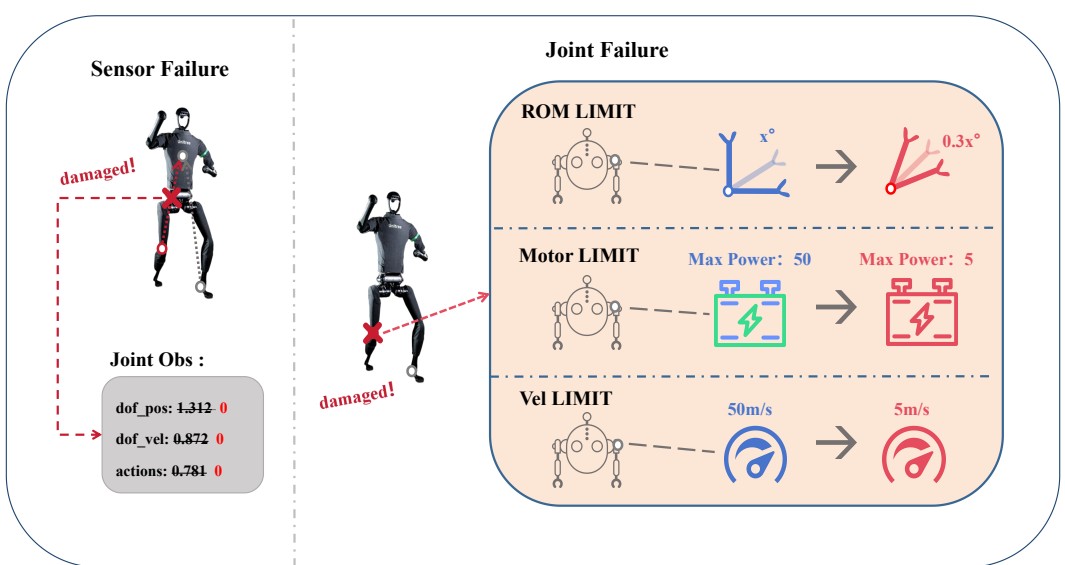

Figure 5: Demonstration of different damage conditions.

Table 10: Malfunction Settings for Training and Inference in the H1 Task.

(a) Training Parameters.

| Parameter | Values |
|---|---|
| Num Envs | 10000 |
| Random Damage Range | [2,4] |
| ROM Limit | Random 30% |
| Motor Limit | 10 |
| Velocity Limit | 5 |
| Random Malfunction Seed | 42 |
| Malfunction Timing | N/A |
| Episode Length | 1000 |

(b) Inference Parameters.

| Parameter | Values |
|---|---|
| Num Envs | 8192 |
| Random Damage Range | [2,3] |
| ROM Limit | Random 30% |
| Motor Limit | 8 |
| Velocity Limit | 3 |
| Random Malfunction Seed | [1, 50, 75] |
| Malfunction Timing | [75, 100, 125] |
| Episode Length | 750 |

## C  MORE EXPERIMENT RESULTS

In this section, we present all experimental results to highlight the overall superiority of our UMC framework across various damage scenarios under different tasks.

In the body of the paper, we have presented the overall performance of all methods across all tasks and damage conditions, so we will not repeat such details here. Instead, we display their perfor-

Table 11: Malfunction Settings for Training and Inference in the Unitree-G1 Task.

(a) Training Parameters.

| Parameter | Values |
|---|---|
| Num Envs | 8192 |
| Random Damage Range | [2,4] |
| ROM Limit | Random 30% |
| Motor Limit for Hip Joints | 8 |
| Motor Limit for Knee Joints | 13 |
| Motor Limit for Ankle Joints | 4 |
| Velocity Limit | 3 |
| Random Malfunction Seed | 42 |
| Episode Length | 1000 |
| Malfunction Timing | N/A |

(b) Inference Parameters.

| Parameter | Values |
|---|---|
| Num Envs | 10000 |
| Random Damage Range | [2,4] |
| ROM Limit | Random 30% |
| Motor Limit for All Joints | 5 |
| Velocity Limit | 3 |
| Random Malfunction Seed | [1, 50, 75] |
| Malfunction Timing | [75, 100, 125] |
| Episode Length | 750 |

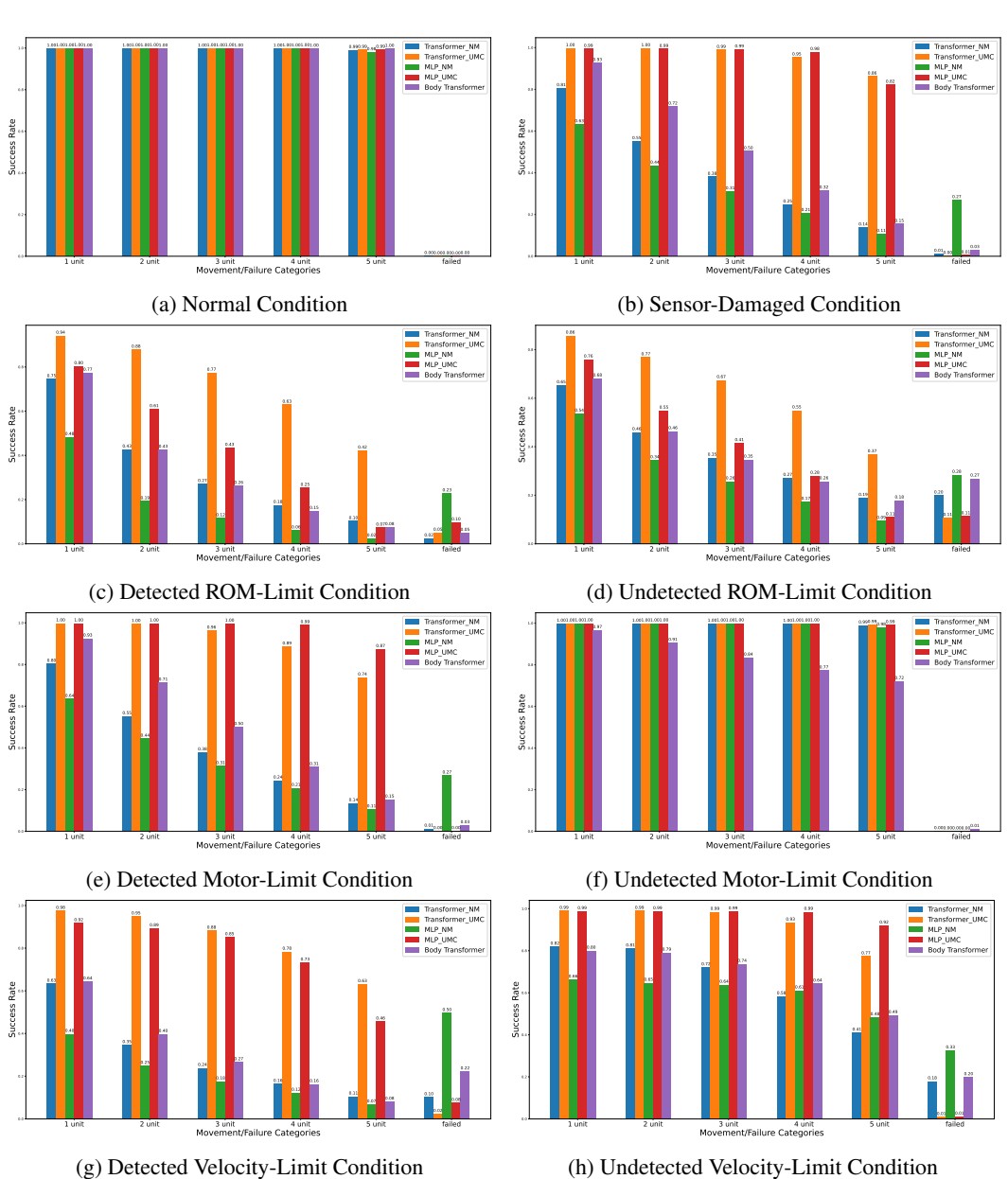

Figure 6: Performance of Five Methods Under Different Damage Conditions in the A1-Walk Task.

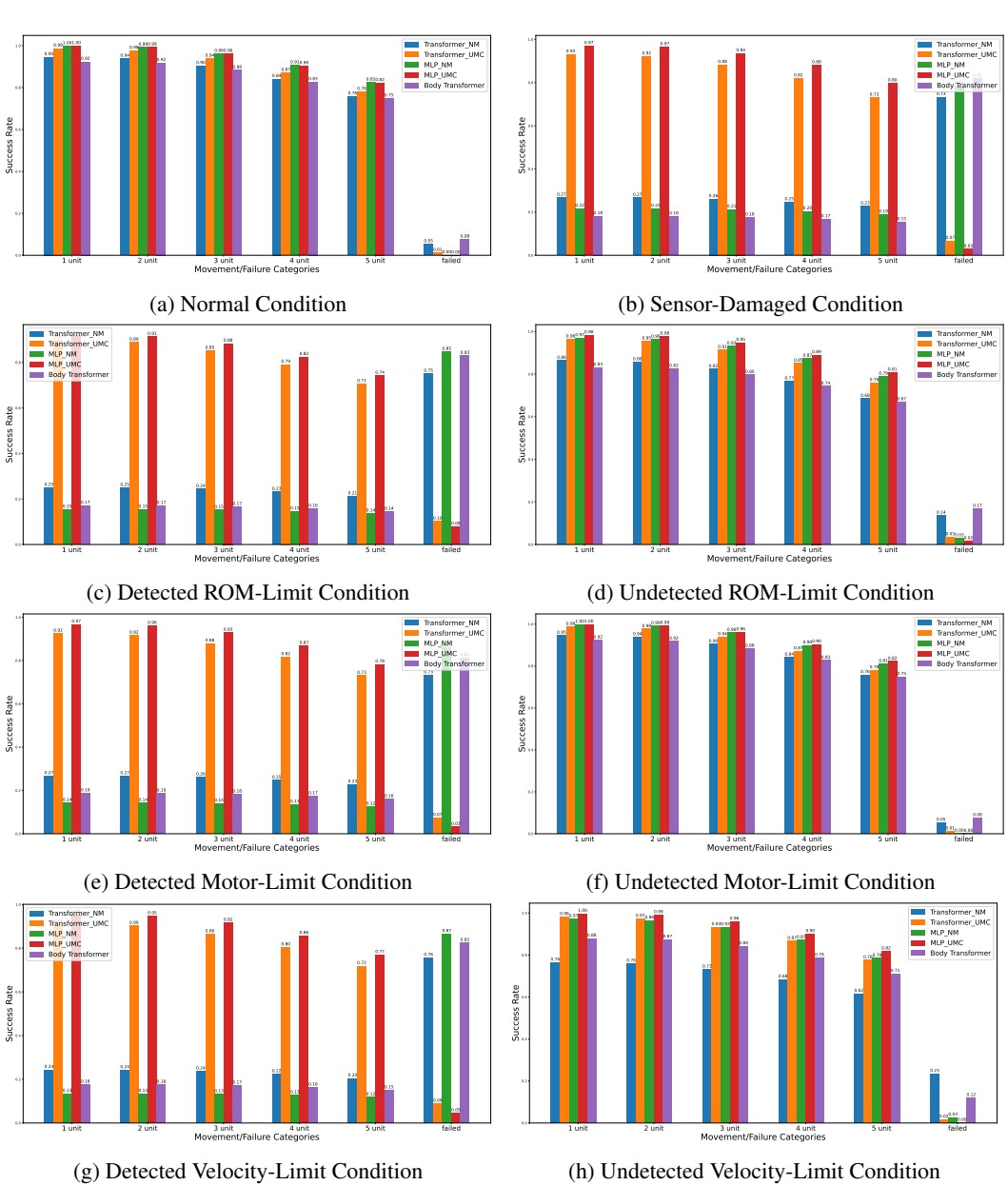

Figure 7: Performance of Five Methods Under Different Damage Conditions in the Unitree-H1 Task.

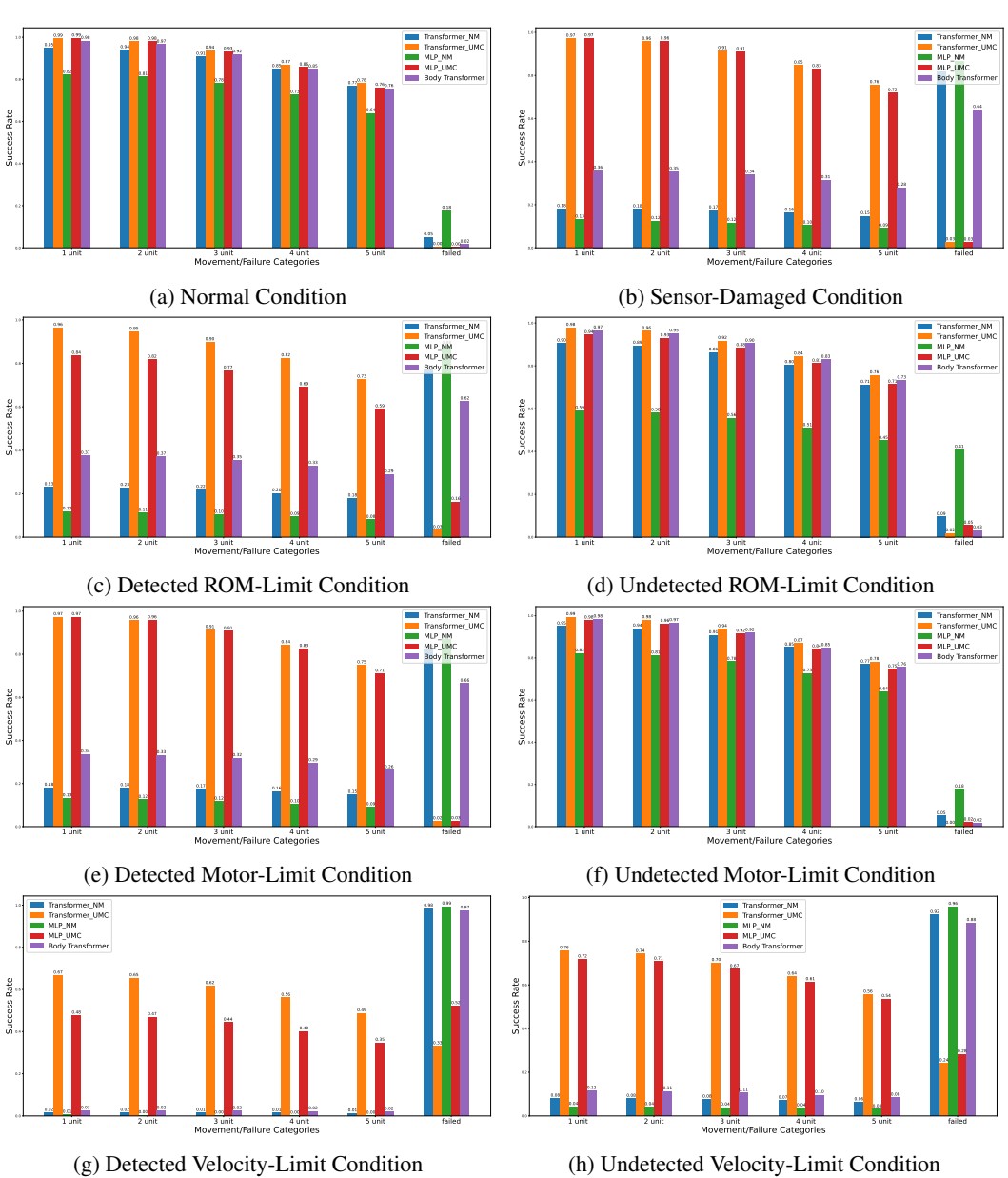

Figure 8: Performance of Five Methods Under Different Damage Conditions in the Unitree-G1 Task.

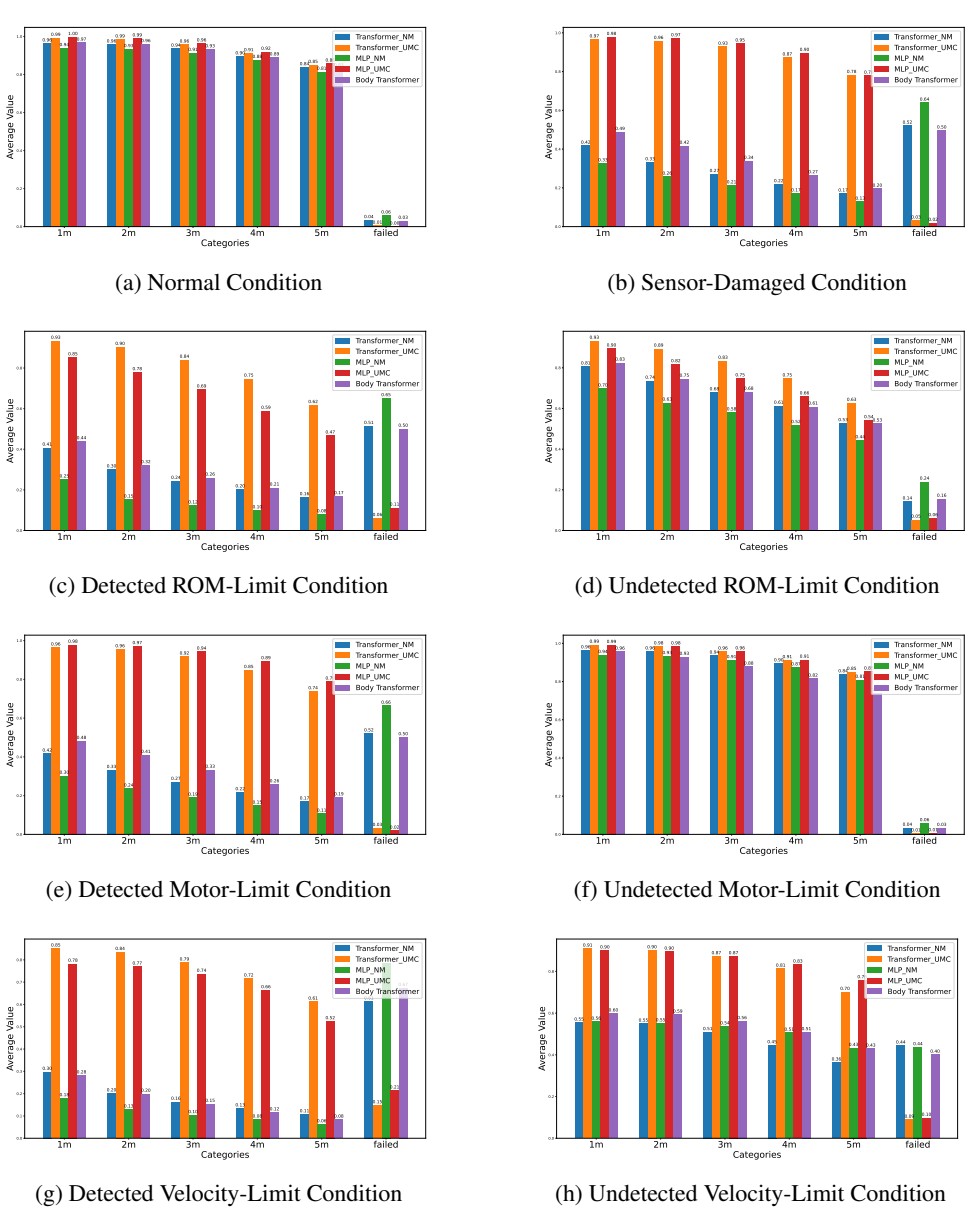

Figure 9: Average Performance of Five Methods Under Different Damage Conditions Across Three Tasks.

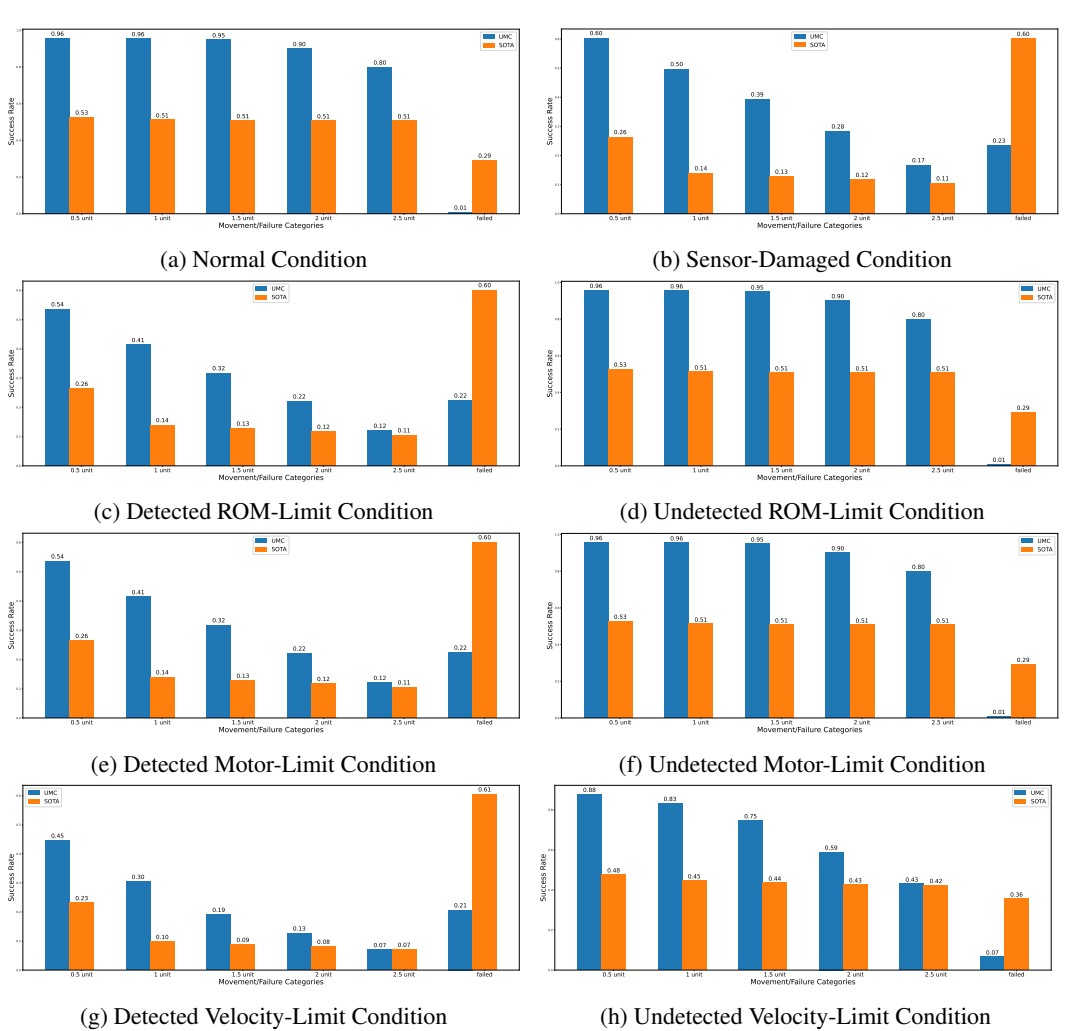

Figure 10: Performance Between UMC and 'MT-FTC' Under Different Damage Conditions in the Solo8 Task.

Table 12: Malfunction Settings for Training and Inference in the Solo8 Task.

(a) Training Parameters.

| Parameter | Values |
| --- | --- |
| Num Envs | 4096 |
| Random Damage Range | [1,3] |
| ROM Limit | Random 30% |
| Motor Limit | 0.75 |
| Velocity Limit | 5 |
| Random Malfunction Seed | 42 |
| Malfunction Timing | N/A |
| Episode Length | 1000 |

(b) Inference Parameters.

| Parameter | Values |
| --- | --- |
| Num Envs | 4096 |
| Random Damage Range | [2,4] |
| ROM Limit | Random 30% |
| Motor Limit for All Joints | 5 |
| Velocity Limit | 3 |
| Random Malfunction Seed | 50 |
| Malfunction Timing | 100 |
| Episode Length | 1000 |

mance under each task's eight damage scenarios. Specifically, Fig. 6 is for the A1-Walk task, Fig. 7 is for the Unitree-H1 task, Fig. 8 is for the Unitree-G1 task, and Fig. 10 is for the Solo8 task (SOTA comparison). Additionally, we calculated the average performance of the three baselines and our two UMC methods across three tasks for each damage condition and show the results in Fig. 9. These statistics demonstrate the superior performance of our UMC framework.

Also, as addressed in the main text, Fig. 6a, Fig. 7a and Fig. 8a illustrate that our UMC framework does not compromise the robot's mobility under normal conditions. On the contrary, as shown in Fig. 8a, UMC even reduces the failure rate of the MLP model by 18% while also enhancing its task-completion performance in the Unitree-G1 task under normal scenarios.

## D   MORE ABLATION DETAILS

Table 13: Inference Parameters for Ablations in section D (except for the stage-count ablation).

| Parameter | Values |
| --- | --- |
| Task | A1 |
| Num Envs | 4000 |
| Random Damage Range | [4,5] |
| ROM Limit | Random 10% |
| Motor Limit | 8 |
| Velocity Limit | 3 |
| Track Width | 6.0 |
| Track Block Length | 6.0 |
| Border Size | 4 |
| Perlin Noise Seed | 100 |
| Random Malfunction Seed | 1 |
| Malfunction Timing | 75 |
| Curriculum Update Threshold | 1 unit |

All ablations, except for the training-stage one, are conducted under the A1 task with one inference damage setting under transformer-based UMC. Table 13 shows the parameter setting during those ablations. We also give the ablation studies on masking value and foundational paradigms of UMC.

**Masking Value.**   We ablate the masking value adopted in our masking mechanism, where the value indicates the observation of the damaged joint. Table 14 shows that zero outperforms the two out-of-distribution values '-100' and '100'. We attribute this to out-of-distribution values exerting greater influence on the model's decision-making. For example, if an action in the observation input is set to 100, though out of range, it still carries some information that the model can analyze. And the impact of such information is greater than that of the default value of 0. Additionally, excessively large values may result in disproportionate rewards or penalties, further affecting the model's performance.

| Values | 1 unit ↑ | 2 unit ↑ | 3 unit ↑ | 4 unit ↑ | 5 unit ↑ | failed ↓ |
|---|---|---|---|---|---|---|
| 100 | 96% | 93% | 87% | 79% | 67% | 3% |
| -100 | 95% | 91% | 86% | 79% | 68% | 4% |
| Default (0) | **97%** | **94%** | **90%** | **84%** | **74%** | **3%** |

Table 14: Average Performance of Transformer-Based UMC with Different Masking Values. The values denote the attention-bias magnitude applied to damaged keys in $M$ (Default $= 0$).

## E  LOSS

The total loss function in PPO adopted in our work is defined as:

$$\mathcal{L} = \mathcal{L}_{\text{surrogate}} + \lambda_1 \cdot \mathcal{L}_{\text{value}} + \lambda_2 \cdot \mathcal{L}_{\text{entropy}}, \tag{7}$$

where $\lambda_1$ and $\lambda_2$ denote weight parameters. $\mathcal{L}_{\text{surrogate}}$ is illustrated in eq. (8), $\mathcal{L}_{\text{value}}$ is illustrated in eq. (10), and $\mathcal{L}_{\text{entropy}}$ is an entropy regularization to encourage exploration.

The actor model in PPO is trained using a clipped surrogate loss to ensure stability in learning. The loss function is defined as:

$$\mathcal{L}_{\text{surro}} = -\mathbb{E}_t \left[ \min \left( r_t(\theta) \cdot A_t, \text{clip}(r_t(\theta), 1 - \epsilon, 1 + \epsilon) \cdot A_t \right) \right], \tag{8}$$

where $t$ represents the timestep index within a trajectory. The ratio $r_t(\theta) = \frac{\pi_\theta(a_t|s_t)}{\pi_{\theta_{\text{old}}}(a_t|s_t)}$ is the probability ratio between the new policy $\pi_\theta(a_t|s_t)$ and the old policy $\pi_{\theta_{\text{old}}}(a_t|s_t)$. The term $A_t$ represents the advantage estimate at timestep $t$, which is computed as:

$$A_t = \sum_{k=0}^{\infty} (\gamma\lambda)^k \delta_{t+k}, \tag{9}$$

where $\delta_{t+k}$ is the temporal difference residual at timestep $t + k$, $\gamma$ is the discount factor, and $\lambda$ balances the bias-variance tradeoff in advantage estimation.

The critic model in UMC shares the same architecture as the actor model without the damage detection module, and it is trained using the value function loss to minimize the error between the predicted value $V(s_t)$ and the target return $R_t$:

$$\mathcal{L}_{\text{value}} = \mathbb{E}_t \big[ \max \big( (V(s_t) - R_t)^2, \\ \text{clip}(V(s_t), V_{\text{old}}(s_t) - \epsilon, V_{\text{old}}(s_t) + \epsilon) - R_t)^2 \big) \big], \tag{10}$$

where $V_{\text{old}}(s_t)$ is the previous value function estimate, and $\epsilon$ is the clipping threshold to ensure stability during training.

## F  LIMITATIONS AND FUTURE WORKS

UMC assumes access to damage indicators produced by an external module $\Gamma$ and does not perform fault detection itself. This modular design is intentional: it decouples fault detection from fault tolerance, letting UMC plug into any detector (rule-based or learned), benefit from detector upgrades without retraining the policy, and simplify safety validation. While UMC remains robust under reasonable misclassification (cf. our noisy-detector ablations), systematic missed detections can limit its effect, and latent faults that never trigger $\Gamma$ remain challenging. Future work includes (i) replacing hard masks with probabilistic/uncertainty-aware masks derived from detector confidence, (ii) joint or alternating training of the detector and UMC to co-adapt, (iii) self-supervised anomaly scoring over multi-modal signals (current–torque mismatch, thermal, IMU) to extend coverage, (iv) hardware studies on latency/jitter and energy-aware or safety-constrained control (e.g., CBF/MPC guidance), and (v) inferring undetectable faults via latent-state belief tracking to further reduce dependence on explicit flags.

## G    LLMs in Paper Writing

Large language models (LLMs, e.g., GPT-4, GPT-5) were used only for refining grammar and sentence structure, with the sole purpose of enhancing readability, clarity, and fluency. They did not contribute to the research ideas, methods, results, or interpretations. All scientific and technical content of this work was conceived, conducted, and written entirely by human authors.

