# OpenReview forum: "UMC: A Unified Approach for Resilient Control of Legged Robots Across Masked Malfunction Training"
_ICLR.cc/2026/Conference — ICLR 2026 Conference Withdrawn Submission_

### Official Review · Reviewer_X7vm · 2025-11-01

**Soundness:** 3
**Presentation:** 2
**Contribution:** 1
**Rating:** 2
**Confidence:** 5

**Summary:**

The paper aims to achieve robust locomotion under diverse conditions of robot malfunction. The authors first define eight damage scenarios that combine (i) sensor status (functional vs damaged) and (ii) three joint damage types (ROM restriction, reduced motor force, limited joint velocity). On top of this taxonomy, they propose a Unified Malfunction Controller (UMC), a two-stage, model-free RL pipeline; 1) Stage I: train a locomotion policy in undamaged environments to get a strong baseline and 2) Stage II: fine-tune the same policy on a mixture of malfunction scenarios (both detectable and undetectable) so that the policy learns to rely on healthy joints and to ignore corrupted signals. The method is shown to work with both transformer-based and MLP-based policies. Experiments on A1, Unitree G1, H1, and Solo8 in IsaacGym/LeggedGym show lower failure rates and longer travel distance after damage than baselines such as BodyTransformer, vanilla transformer, and an MT-FTC variant.

**Strengths:**

1. Clear problem framing and taxonomy.
* The paper usefully enumerates eight realistic malfunction cases (sensor OK/broken x three joint degradations). This is more systematic than many “one leg removed” or “one motor stuck” toy setups. It makes the contribution easy to situate.

2. Broad evaluation on multiple robots and multiple damage timings.
* They train/test on multiple robots in multiple damage configurations: A1, H1, G1, Solo8, plus different damage times and combinations, and they evaluate success at increasing travel radii (1–5 units).

**Weaknesses:**

1. Missing the most relevant 2023–2025 fault-/damage-tolerant quadruped literature and novelty over the existing works.
* The related work almost entirely stays in (i) generic legged RL (RMA, BodyTransformer, parkour / LeggedGym), (ii) transformer in robotics, and (iii) self-recovering. But it does not cite or compare to exactly the line of works that study fault-tolerant / damage-aware locomotion with explicit joint failures, such as the following list, just to name a few. These recent papers focus heavily on fault-tolerant learning policies and fault recovery strategies. The lack of comparison or discussion with these works is a critical oversight.
For instance, ICRA 2024 [2] proposed a random joint masking for quadrupedal robots in a way that’s directly related to this work, also utilizing a two-stage training method, similar to this paper.

* [1] Ft-net: Learning failure recovery and fault-tolerant locomotion for quadruped robots, RA-L 23
* [2] Learning Quadrupedal Locomotion with Impaired Joints Using Random Joint Masking, ICRA 24
* [3] Meta Reinforcement Learning of Locomotion Policy for Quadruped Robots With Motor Stuck, T-ASE 24
* [4] Multi-Task Learning of Active Fault-Tolerant Controller for Leg Failures in Quadruped robots, ICRA 24
* [5] FT-CPG: Learning Central Pattern Generators for Fault-Tolerant Quadruped Locomotion Under Multi-Joint Failures, RA-L 25
* [6] DreamFLEX: Learning Fault-Aware Quadrupedal Locomotion Controller for Anomaly Situation in Rough Terrains, ICRA 25
* [7] Fault Joint Detection and Adaptive Fault-Tolerant Control of Legged Robots Under Joint Partial Failures, RA-L 25

2. Missing comparison with the above works.
* The paper would be much stronger if it included direct comparative experiments, benchmarking against the fault-tolerant models (i.e., the above-listed works). Without these, it is hard to determine if UMC is genuinely an improvement over recent solutions, or if it merely offers a rebranding of existing approaches.

3. Missing real-world experiments.
* Another critical difference with the above-listed works and the current paper is that the previous works all include real-world demonstrations that show each proposed method is valid in real-world robots. The paper also needs to include at least one demonstration.

**Questions:**

Please see the weakness parts.

---

### Official Review · Reviewer_AyYD · 2025-11-01

**Soundness:** 3
**Presentation:** 3
**Contribution:** 2
**Rating:** 4
**Confidence:** 3

**Summary:**

This paper introduces the Unified Malfunction Controller (UMC), a novel two-stage reinforcement learning framework designed to enhance the robustness of legged robots under a wide range of joint and sensor malfunctions. The authors systematically categorize eight damage scenarios based on sensor status (functional/damaged) and joint impairment type (ROM restriction, reduced motor force, limited velocity). UMC employs a two-stage training pipeline: Stage I trains the policy under normal conditions to establish a strong baseline, and Stage II fine-tunes it with a mixture of malfunction scenarios, including both detectable and undetectable faults. For detectable malfunctions, a masking strategy is introduced to explicitly suppress corrupted sensor inputs. The method is architecture-agnostic and evaluated on both Transformer and MLP backbones across multiple locomotion tasks (A1, H1, G1, Solo8). Extensive experiments show that UMC significantly reduces failure rates and improves task completion under diverse damage conditions, outperforming several strong baselines.

**Strengths:**

1.	The paper provides a well-structured and comprehensive taxonomy of eight malfunction scenarios, covering both sensor and joint failures. This systematic approach strengthens the evaluation and ensures broad applicability to real-world conditions.
2.	The masking mechanism for detectable malfunctions is a key contribution. By explicitly filtering out faulty sensor inputs, the policy can focus on reliable signals, leading to more resilient behavior.

**Weaknesses:**

1. The idea behind this paper is very good—addressing how to maintain balance when a robot's joints are partially damaged. However, the definition of "damage" in the paper is not sufficiently detailed or logical. For instance, stating that sensors "fail to provide correct observation and output only zeros (thus detectable)" seems more like a communication failure rather than actual physical damage to the robot.
2. The description of the tasks used to evaluate the model's performance is incomplete. Based on the figures in the paper, it appears that only locomotion training for the lower-body joints was conducted, with the upper body fixed. However, this point is not explicitly mentioned in the text, which does not align with the contributions claimed.
3. The paper lacks real-robot demonstrations and qualitative examples. While physically damaging a real robot may be costly, adding noise to specific motors could simulate such scenarios. Additionally, the examples provided in the simulation are not sufficiently convincing.

**Questions:**

see weakness

---

### Official Review · Reviewer_4vVC · 2025-11-01

**Soundness:** 2
**Presentation:** 3
**Contribution:** 2
**Rating:** 4
**Confidence:** 5

**Summary:**

The paper proposes UMC, a two-stage training pipeline to obtain a single locomotion policy that can handle a wide range of joint/sensor malfunctions in simulation on several legged robot models. Stage 1 trains a PPO locomotion policy on normal, undamaged environments. Stage 2 fine-tunes this policy on a balanced mixture of 8 manually defined malfunction scenarios that cover (i) detectable faults and (ii) undetectable faults. For detectable faults, the policy uses an attention-level masking mechanism so damaged joints do not influence healthy joints. Experiments in Isaac Gym / Legged Gym show consistent gains over standard transformer/MLP policies that are not trained on such a mixture.

**Strengths:**

1. Clear and practical problem setup: the paper makes a useful distinction between detectable and undetectable malfunctions, which does reflect how robots fail in practice.
2. Systematic malfunction taxonomy: the 8-case construction (normal, three undetectable actuation degradations, and four “sensor damaged” cases) gives a concrete recipe others can reimplement.
3. Simple training recipe: pretrain on clean, then fine-tune on a balanced damage mixture; this is easy to adopt in existing legged-RL codebases.
4. Attention-level masking for detectable damage: injecting the mask at the attention-logit level is a clean way to stop damaged joints from affecting the rest of the body.
5. Broad simulated evaluation: shown on multiple robot morphologies (quadruped and humanoid-like), which supports the “unified” claim within simulation.

**Weaknesses:**

1. The biggest weakness is the complete absence of real-robot validation. The paper is motivated by real hardware failures (broken sensors, weakened joints), but all results are in simulation, and there is no evidence that the proposed 8-case taxonomy and attention masking will transfer to real sensing/actuation noise or real-time constraints.
2. The malfunction space is still hand-enumerated. Although the 8 scenarios are reasonable, there is no analysis of coverage (e.g., multiple simultaneous joint degradations, contact-related faults, time-varying degradation).
3. The paper would benefit from a deeper analysis of when the policy fails in undetectable cases (where the sensor does not tell you which joint is bad), since those are the closest to real intermittent wear/faults.
4. Transition-time robustness (fault happens mid-episode) is not explored in detail.
5. The comparison to other robust/privileged-/RMA-style training pipelines could be sharper, especially to clarify when a plain “train on lots of damaged variants” baseline stops working.

**Questions:**

1. Maybe try any zero-shot hardware rollout (even slow walking) to see whether the attention masking causes unsafe actions on real actuators?
2. Can the masking handle multiple simultaneous detectable joint failures without retraining?
3. How sensitive is performance to the damage mixing ratio in Stage 2?
4. For undetectable cases, did you consider adding a small latent “fault” token to help the policy internally infer the faulty joint?

---

### Official Review · Reviewer_BaMK · 2025-11-02

**Soundness:** 3
**Presentation:** 3
**Contribution:** 1
**Rating:** 2
**Confidence:** 5

**Summary:**

This paper proposes a 2-stage training procedure -- UMC -- to address the problem of sensor failures and joint malfunctions for legged robots. In the first stage of training, UMC trains the agent on no failures as a pre-training step, and in the second stage, they incorporate a masking mechanism wherein they make the robots robust to failed sensors/joints. This paper also classifies the failures into 8 categories. Experiments on simulated benchmarks show that the proposed pipeline is robust to failures across the spectrum of the 8 categories.

**Strengths:**

The categorization of failures into different categories is a useful and novel contribution when it comes to systematically investigating the resiliency of locomotion policies.

**Weaknesses:**

**Disclaimer**: I've reviewed this paper in a prior conference, and part of my review would be considering the comments that I'd raised in a previous version of the paper.


1. My main concern that was raised in the earlier review still remains -- the originality/novelty of this work. Authors do point out poor performance of [1] on L60-61, which is a very similar method; however, it is unclear which method in Fig. 1(b) corresponds to [1]. Further, the UMC-MLP method for locomotion is very similar to [1] for locomotion, where the authors mask out the damaged sensors with a binary mask. I'd like the authors to elaborate on what aspect this is different from [1].

2. Discussing LocoFormer [2], which is another work close to training robust policies in the related work section, would be beneficial.


3. Lack of any sim2real experiments makes it further less novel compared to [2, 1].

------
**References**

[1] Simple Masked Training Strategies Yield Control Policies That Are Robust to Sensor Failure, Skand Peri et al., CoRL 2024

[2] LocoFormer: Generalist Locomotion via Long-Context Adaptation, Min Liu et al., CoRL 2025

**Questions:**

1. It appears that the SMS approach from the baselines is not shown in Table 2, even though the authors claim it on L421. I would assume that MLP-UMC is similar to RME [1] and would like the authors to expand on that if it's not the case.

2. For each result reported in Tables 2-6, how many seeds were used to train these models? There's no standard deviation provided.

---

### Note · Authors · 2025-11-16

I have read and agree with the venue's withdrawal policy on behalf of myself and my co-authors.